# Effects of Cardiac Contractility Modulation Electrodes on Tricuspid Regurgitation in Patients with Heart Failure with Reduced Ejection Fraction: A Pilot Study

**DOI:** 10.3390/jcm11247442

**Published:** 2022-12-15

**Authors:** Daniele Masarone, Michelle M. Kittleson, Stefano De Vivo, Antonio D’Onofrio, Ishu Rao, Ernesto Ammendola, Vittoria Errigo, Maria L. Martucci, Gerardo Nigro, Giuseppe Pacileo

**Affiliations:** 1Heart Failure Unit, Department of Cardiology, AORN dei Colli Monaldi Hospital, 80131 Naples, Italy; 2Cedars-Sinai Heart Institute, Los Angeles, CA 90048, USA; 3Electrophysiology and Cardiac Pacing Unit, Department of Cardiology, AORN dei Colli Monaldi Hospital, 80131 Naples, Italy; 4Impulse Dynamics, Inc., Marlton, NJ 08701, USA; 5Cardiology Unit, Department of Medical Translational Sciences, Monaldi Hospital, University of Campania “Luigi Vanvitelli”, 80138 Naples, Italy

**Keywords:** cardiac contractility modulation, optimizer smart, heart failure reduced ejection fraction, tricuspid regurgitation

## Abstract

Background: Cardiac contractility modulation (CCM) is an innovative therapy for heart failure with reduced ejection fraction delivered by a cardiac implantable device (Optimizer Smart^®^). One of the most prominent periprocedural complications common to all cardiac implantable devices (CIDs) is tricuspid regurgitation (TR) due to the placement of the right ventricular endocardial leads. To date, no published studies have assessed the changes in the TR degree in patients with heart failure with reduced ejection fraction (HFrEF) who received an implantable cardioverter-defibrillator (ICD) after the implantation of cardiac contractility modulation therapy devices. Objective: This study aimed to evaluate the effect of the implantation of the trans-tricuspid leads required to deliver CCM therapy on the severity of TR in patients with HFrEF who previously underwent ICD implantation. Methods: We enrolled 30 HFrEF patients who underwent CCM therapy between November 2020 and October 2021. For all the patients, echocardiographic evaluations of TR were performed according to current guidelines 24 h before and six months after the Optimizer Smart^®^ implant was applied. Results: At the 6-month follow-up, the grade of TR remained unchanged compared to the preimplant grade. The value of the vena contracta (VC) of TR was 0.40 ± 0.19 cm in the preimplant period and 0.45 ± 0.21 cm at the 6-month follow-up (*p* = 0.33). Similarly, the TR proximal isovelocity surface area (PISA) radius value was unchanged at follow-up (0.54 ± 0.22 cm vs. 0.62 ± 0.20 cm; *p* = 0.18). No statistically significant difference existed between the preimplant VC and PISA radius values, irrespective of the device type. Conclusions: The implantation of right ventricular electrodes for the delivery of CCM therapy did not worsen tricuspid regurgitation in patients with HFrEF and ICD.

## 1. Introduction

Cardiac contractility modulation (CCM) is an innovative therapy for the treatment of heart failure (HF) with mildly reduced (HFmrEF) and reduced ejection fraction (HFrEF) that modulates the myocardial contraction force through the delivery of non-excitatory impulses [1]. The device used for CCM therapy delivery, the Optimizer Smart^®^, generates high-amplitude (from 4.0 V to 7.5 V) biphasic electrical signals during the absolute refractory period of the cardiac cycle, leading to an improvement in calcium handling and, consequently, in cardiac contractility and performance [2].

In patients with HFmrEF and HFrEF, CCM therapy improves the symptoms and quality of life, reduces the number of hospitalizations, and promotes biventricular reverse remodeling [3]. However, Optimizer Smart implantation is an invasive procedure that is potentially subject to several theorized early and late complications. It should be noted that the actual 30-day significant adverse event (SAE) rate of Optimizer implantation is similar to that of dual-chamber pacemaker implantation (8.8% vs. 9.1%, respectively) [4,5].

One of the most prominent periprocedural complications common to all cardiac implantable devices (CIDs) is tricuspid regurgitation (TR) due to the placement of the right ventricular endocardial leads [6].

The mechanisms through which endocardial leads can result in TR are numerous and can be characterized as structural [7] (due to valve deformity from the impingement of the leads to the valve leaflet or valve perforation), functional [8] (recurrent embolization from lead thrombosis, resulting in pulmonary hypertension and TR secondary to right ventricular dilatation), or physiologic [9] (due to TR resulting from the RV-pacing-induced worsening of HF).

However, more than 650 patients with HFmrEF and HFrEF who participated in two randomized controlled trials and a large CCM therapy registry had Optimizer Smart^®^ devices and concomitant implantable defibrillators fitted (with at least three leads crossing the tricuspid valve), and there were no reported cases of worsening TR [3,4,10].

As no specific published studies exist, the purpose of this study was to evaluate the effect of the implantation of the trans-tricuspid leads required to deliver CCM therapy on the severity of TR in patients with HFrEF who had previously received implantable cardioverter-defibrillator (ICD) implants by echocardiography.

## 2. Methods

### 2.1. Study Population

We prospectively and consecutively enrolled all the patients diagnosed with HFrEF who underwent CCM therapy between November 2020 and December 2021 according to the European Society of Cardiology guidelines. 

The following inclusion criteria were used: −Left ventricular ejection fraction of <40%;−New York Heart Association (NYHA) class II–III;−Referral for CCM implant due to the >2 unplanned visits or hospitalization in the last 12 months and/or the persistence of HF-related symptoms despite the use of optimal medical therapy;−A QRS duration of <120 msec.

The following exclusion criteria were used:−Acute coronary syndrome in the previous three months;−ICD implantation in the previous twelve months;−Severe tricuspid regurgitation (i.e., vena contracta >7 mm, proximal isosurface radius >9 mm). 

The demographic, clinical, and laboratory data were acquired from stable patients 24 h before the Optimizer Smart^®^ implantation.

The study was conducted according to the Declaration of Helsinki. For all the patients, signed informed consent was obtained, and approval was received from the institutional review board of AORN dei Colli-Ospedale Monaldi (deliberation No. 903/2020).

### 2.2. Echocardiographic Evaluation

Standard transthoracic echocardiography and Doppler evaluation were performed using commercially available equipment (Vivid E9, GE Healthcare, Milwaukee, WI, USA) according to the international guidelines [11,12]. 

Two independent observers, who were blinded to the clinical details of the patients enrolled, analyzed all the echocardiographic studies, and an average of 3–5 cardiac cycles were performed for each parameter.

According to the international recommendations, TR was assessed by color Doppler evaluation in the apical four-chamber view [13]. The degree of TR was classified as mild in the presence of a vena contracta (VC) <0.3 cm and proximal isovelocity surface area (PISA) radius <0.5 cm, as moderate with a VC >0.3 cm and <0.7 cm and PISA radius >0.5 cm and <0.9 cm, and as severe with a VC >0.7 cm and PISA radius >0.9 cm.

For all the patients, echocardiographic evaluations were performed 24 h before and six months after the Optimizer Smart^®^ implant was applied.

### 2.3. Optimizer Smart^®^ Implant

The implantation procedure of the Smart Optimizer (Impulse Dynamics Inc., Marlton, NJ, USA) was performed after the patient’s sedation under local anesthesia. 

Two electrodes, which are necessary for detecting ventricular activity and the subsequent CCM therapy delivery, were attached to the right side of the interventricular septum through the right subclavian vein. These leads were then connected to the Optimizer Smart, and the device was implanted in a subcutaneous pocket with the charging coil facing in the anterior direction.

### 2.4. Statistical Analysis

Statistical analyses were performed using Prism 9 (GraphPad Software, San Diego, CA, USA). The demographic and clinical variables were expressed as the mean ± standard deviation. The categorical variables were expressed as numbers and percentages. Differences between the baseline and treatment values were compared using Wilcoxon’s rank test for a non-normal distribution and the paired t-test for a normal distribution. 

Receiver operating characteristic curve analysis was performed to select the optimal cut-off values for the echocardiographic measurements. In addition, the reproducibility of the measurements was determined in the case of all the patients. The inter-observer and intra-observer variability of the echocardiographic measures were examined using Pearson’s two-tailed bivariate correlations and Bland–Altman analysis. Correlation coefficients, 95% confidence limits, and percentage errors were reported.

All the *p*-values were two-sided, and *p* < 0.05 indicated statistical significance.

## 3. Results

Thirty-two patients with HfrEF underwent Optimizer Smart^®^ implantation during the study period. Of these, two patients has severe tricuspid regurgitation; thus, they were not enrolled in the study. 

The demographic, clinical, and echocardiographic characteristics of the 30 patients enrolled in the study are presented in Table 1.

All the patients had a previously implanted device. In total, 14 (46.7%) had a dual-chamber implantable cardioverter-defibrillator (ICD-DR), 12 (40%) had a device for cardiac resynchronization therapy with defibrillation back-up (CRT-D), and 4 (13.3%) had a single-chamber implantable cardioverter-defibrillator (ICD-VR). 

The r coefficients for Pearson’s two-tailed bivariate correlations were 0.92 and 0.87 according to the Bland–Altman analysis.

At the 6-month follow-up, the grade of TR remained unchanged compared to the pre-implant grade (Figure 1). The VC value of TR was 0.40 ± 0.19 cm in the pre-implant period and 0.45 ± 0.21 cm at 6 months (*p* = 0.33). Similarly, the PISA radius value of TR was unchanged at follow-up (0.54 ± 0.22 cm vs. 0.62 ± 0.20 cm; *p* = 0.18). No statistically significant difference existed between the pre-implant VC and PISA radius values, irrespective of the device type (Figure 2A,B). 

In addition, as shown in Table 2 and Figure 3, at the six-month follow-up, the CCM therapy induced right ventricular reverse remodeling and reduced systolic pulmonary pressure values.

These data confirm the absence of hemodynamically significant worsening of tricuspid regurgitation after the implantation of the electrodes that deliver CCM therapy. In the multivariable analysis (Table 3), the non-significant worsening of TR was associated with left ventricular ejection fraction and pulmonary artery systolic pressure.

## 4. Discussion

The main findings of this study are as follows:(1)The implantation of pacemaker leads to deliver CCM therapy did not result in a worsening of TR at six months compared with the baseline.(2)The absence of TR worsening was independent of the type of CIED previously implanted and, therefore, the presence and number of endocardial leads already implanted.

With the advent of CIED-based therapies for HfrEF, several investigations have shown a worsening of TR associated with the implantation of an ICD and CRT-D [14]. 

However, to the best of our knowledge, no prospective study in the literature has evaluated the effects of implanting endocardial leads for CCM therapy delivery on tricuspid valve function.

Our results demonstrate, for the first time, that CCM implantation does not increase the severity of TR. This finding is particularly important, because most patients enrolled in the study had a pre-existing CIED with one or more endocardial leads.

In our population, adding two leads on the right side of the interventricular septum did not worsen the extent of TR in the patients with a CRT-D, ICD-DR, or ICD-VR.

After the implantation of the CIED leads, the worsening of TR (described in 16–25% of the patients) could develop or worsen because of several proposed mechanisms, including the physical impingement of the lead on the valve [15], fibrous tissue formation on the valve leaflets [16], and, rarely, the perforation and entrapment of the lead in the valve apparatus. Additionally, and perhaps most commonly, RV-pacing-induced worsening heart failure worsens the RV hemodynamics, resulting in the worsening of TR [17].

Lead implantation for CCM therapy is not associated with any of these mechanisms for various reasons. Firstly, though both leads used for CCM delivery are placed in the right ventricle, CCM does not result in the pacing of the heart (that is, no excitatory event is produced), so that the normal depolarization patterns remain unaltered.

Thus, RV-pacing-induced HF and the subsequent deterioration of tricuspid valve function do not occur.

Additionally, CCM improves left ventricular systolic and diastolic function and induces left ventricular remodeling [18]. These effects could be protective against heart-failure-induced TR [19]. After reducing the LV filling pressures, the tricuspid valve pressure gradient is similarly decreased, and TR is not likely to worsen but may, indeed, improve [20]. 

In addition, in our population, CCM improved the right ventricle function and right ventricular–pulmonary artery coupling. These effects are mediated by both the improvement in myocardial contractility and reduction in pulmonary artery systolic pressure (PASP) [21].

In our study, the Δ of TR at six months was associated with left ventricular ejection fraction (LVEF) and pulmonary artery systolic pressure (PASP).

A previous study showed that in patients with HfrEF, the degree of TR was correlated with left ventricular ejection fraction (LVEF), which is associated with the tethering of the leaflet of tricuspid valve [22]. This fact suggests that ventricular interdependence plays a significant role in determining tricuspid valve competence, presumably via the effect of left ventricular dysfunction on the interventricular septum, to which the septal leaflet of the tricuspid valve is attached [23]. 

PASP is one of the main determinants of TR in patients with HfrEF who have previously received CIEDs implants [24]. However, it appears that the remodeling of the right heart in response to elevated pulmonary pressure, and not only the increase in PASP, represents the major mechanism responsible for TR in these patients [25]. 

Thus, the increase in LVEF and reduction in PASP induced by CCM could represent the main mechanisms that contribute to the “neutral” effects that the implantation of CCM electrodes have on the degree of TR in patients with HFrEF. 

## 5. Study Limitations

Our study had some limitations. Firstly, we used 2D echocardiography to assess the TR and RV function, whereas 3D echocardiography might have been more accurate in quantifying TR. However, we used more quantitative parameters determined by the guidelines to optimally evaluate TR in standard clinical practice.

Secondly, despite the prospective nature of our study, our study population was small. Therefore, our results will need to be confirmed in a larger population. Finally, the endpoints were assessed at six months post-implantation, and it has yet to be seen what long-term effects may occur in the future. In particular, the fibrotic scarring of leads of the tricuspid apparatus may take years to develop and manifest effects.

## 6. Conclusions

In this pilot study, the implantation of right ventricular electrodes for the delivery of CCM therapy did not appear to worsen TR in patients with HFrEF who had previously undergone CIED implantation. The mechanisms that may prove to be particularly important are (1) the lack of RV pacing through CCM delivery (thus preventing RV-pacing-induced heart failure) and (2) biventricular function improvement, which secondarily improves the RV hemodynamics. HF specialists and electrophysiologists should be aware of this so as to avoid depriving patients of a safe and effective therapy for HFrEF due to the fear of worsening TR.

## Figures and Tables

**Figure 1 jcm-11-07442-f001:**
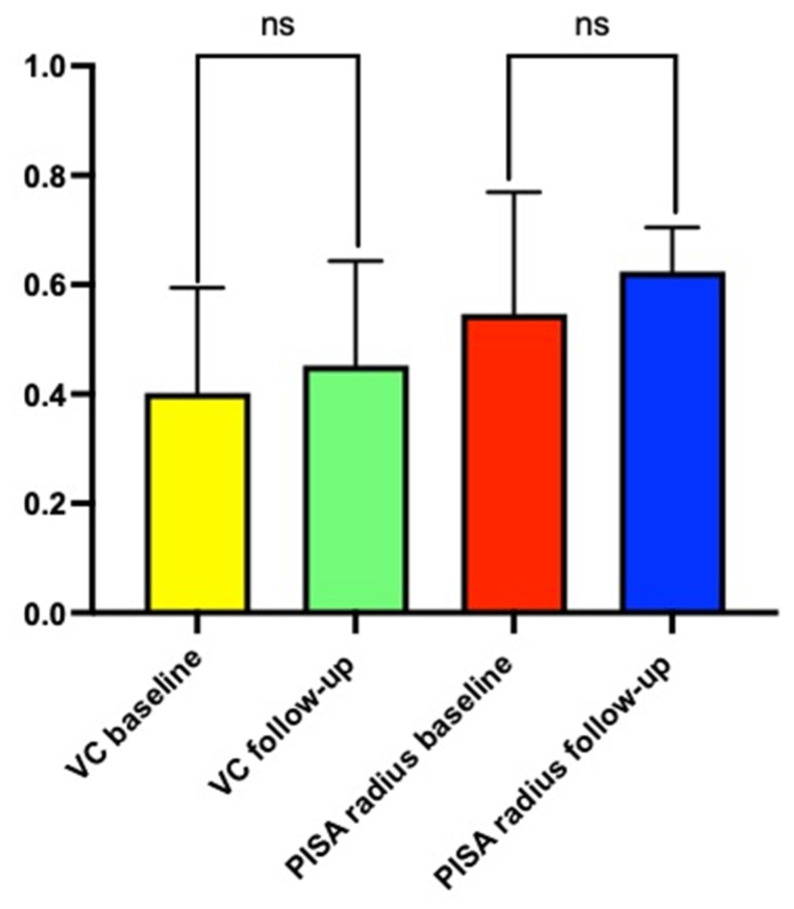
Value of vena contracta and proximal isosurface area radius of tricuspid regurgitation before and after Optimizer^®^ Smart implantation. VC: vena contracta; PISA: proximal isosurface area radius. ns: non-significance.

**Figure 2 jcm-11-07442-f002:**
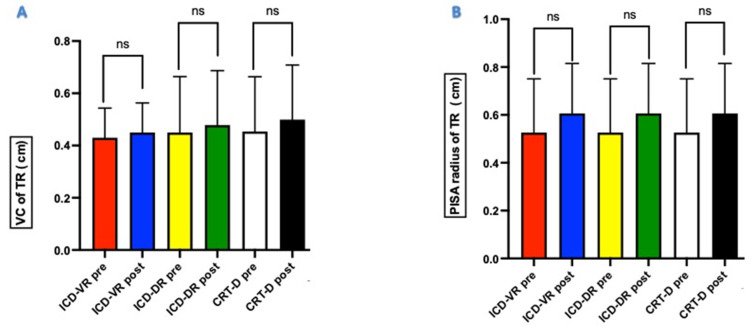
Value of vena contract (panel (**A**)) and proximal isosurface area radius (panel (**B**)) of tricuspid regurgitation before and after Optimizer^®^ Smart implantation. ns: non-significance. ICD-VR: single-chamber implantable cardioverter-defibrillator; ICD-DR: dual-chamber implantable cardioverter-defibrillator; CRT-D: cardiac resynchronization therapy with implantable cardioverter-defibrillator back-up.

**Figure 3 jcm-11-07442-f003:**
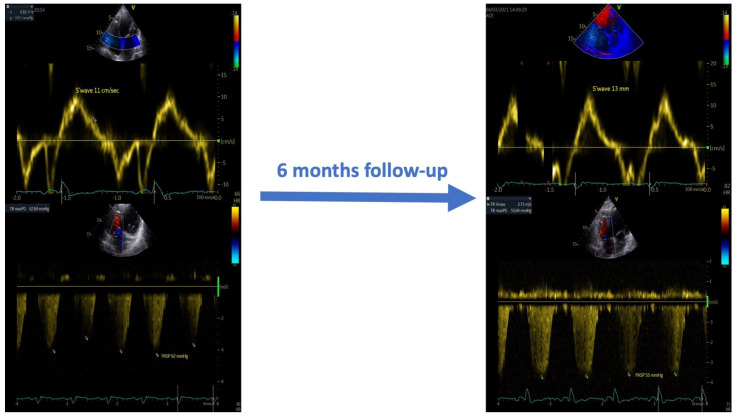
Effects of CCM therapy on the systolic function and pulmonary artery systolic pressure at the six-month follow-up.

**Table 1 jcm-11-07442-t001:** Demographic, clinical, and echocardiographic characteristics of the study population.

Variable	Total Population (n = 30)
Age (mean ± SD)	59.5 ± 12.9 years
Female sex (n, %)	5 (16.6%)
Ischemic etiology (n, %)	13 (43.3%)
Hypertension (n, %)	10 (33.3%)
Diabetes (n, %)	8 (26.6%)
NYHA class II (n, %)	6 (20%)
NYHA class III (n, %)	24 (80%)
SBP (mean ± SD)	108 ± 12.3 mmHg
DBP (mean ± SD)	62 ± 5.7 mmHg
HR (mean ± SD)	62 ± 10.2 b/m
NT-pro-BNP (mean ± SD)	3956 ± 2872 pg/mL
Atrial fibrillation	12 (40%)
ICD-DR	14 (46.6%)
ICD-VR	2 (6.6%)
S-ICD	2 (6.6%)
CRT-D	12 (40%)
Hb (mean ± SD)	11.3 ± 1.2 g/dL
Creatinine (mean ± SD)	1.2 ± 0.7 mg/d:
e-GFR (mean ± SD)	45.9 ± 13.6 mL/min/1.73 m²
LVEDV (mean ± SD)	225.8 ± 51.6 mL
LVESV (mean ± SD)	162.4 ± 41.8 mL
LVEF (mean ± SD)	30.5 ± 3.6%
E wave (mean ± SD)	110.5 ± 38.7 cm/sec
E’ average (mean ± SD)	5.8 ± 3.2 cm/sec
E/e’ average (mean ± SD)	15.5 ± 4.2
DecT (mean ± SD)	142.8 ± 45.3 m/sec
LAVi (mean ± SD)	47.3 ± 11.5 mL/m^2^
RVOT prox (mean ± SD)	28.7 ± 4.2 mm
RVOT dist (mean ± SD)	25.3 ± 3.8 mm
RVD 1 (mean ± SD)	29.2 ± 4.8 mm
RVD 2 (mean ± SD)	27.5 ± 5.2 mm
RVD3 (mean ± SD)	63.4 ± 6.2 mm
TAPSE (mean ± SD)	13.6 ± 5.6 mm
S wave (mean ± SD)	10.3 ± 1.5 cm/sec
PASP (mean ± SD)	37.6 ± 8.2 mmHg
TR mild (n, %)	17 (56.6%)
TR moderate (n, %)	13 (43.4%)

NYHA: New York Heart Association; SBP: systolic blood pressure; DBP: diastolic blood pressure; HR: heart rate, NT-pro-BNP: NT-pro-brain natriuretic peptides; ICD-DR: implantable cardioverter-defibrillator dual chamber; ICD-VR: implantable cardioverter-defibrillator single chamber; S-ICD: subcutaneous implantable cardioverter-defibrillator; CRT-D: cardiac resynchronization therapy with defibrillator back-up; Hb: hemoglobin; e-GFR: estimated glomerular filtration rate; LVEDV: left ventricular end-diastolic volume; LVESV: left ventricular end-systolic volume; LVEF: left ventricular ejection fraction; E wave: peak early mitral inflow velocity; e′ average: average septal and lateral peak early diastolic mitral annular velocity; DecT: deceleration time; LAVi: left atrium volume index; RVOT prox.: right ventricle outflow tract dimension on the proximal sub-valvular level; RVOT distal: right ventricle outflow tract dimension on the distal or pulmonic valve level; RVD1: right ventricle basal dimension; RVD2: right ventricle mid-cavity dimension; RVD3: right ventricle longitudinal dimension; TAPSE: tricuspid annular plane systolic excursion; S wave: peak systolic of the free wall of the right ventricle; PASP: pulmonary artery systolic pressure; TR: tricuspid regurgitation.

**Table 2 jcm-11-07442-t002:** Effects of CCM on the right ventricular dimensions, systolic function, and hemodynamic parameters at the six-month follow-up.

Parameter	Baseline	6-Month Follow-Up	*p*-Value
RVOT prox (mean ± SD)	28.7 ± 4.2 mm	26.3 ± 3.8	0.042
RVOT dist (mean ± SD)	25.3 ± 3.8 mm	22.9 ± 4.5 mm	0.037
RVD 1 (mean ± SD)	29.2 ± 4.8 mm	27.2 ± 3.3 mm	0.026
RVD 2 (mean ± SD)	27.5 ± 5.2 mm	26.2 ± 4.8 mm	0.022
RVD3 (mean ± SD)	63.4 ± 6.2 mm	61.9 ± 3.8 mm	0.031
TAPSE (mean ± SD)	13.6 ± 5.6 mm	16.7 ± 4.6 mm	0.012
S wave (mean ± SD)	10.3 ± 1.5 cm/s	12.3 ± 2.8 cm/s	0.017
PASP (mean ± SD)	37.6 ± 8.2 mmHg	33.6 ± 4.7 mmHg	0.035
PAMP	20.3 ± 7.5 mmHg	15.8 ± 4.2 mmHg	0.043
PCWP	12.6 ± 5.8 mmHg	9.3 ± 2.9 mmHg	0.031

RVOT prox.: right ventricular outflow tract proximal diameter; RVOT dist.: right ventricular outflow tract distal diameter; RVD 1: right ventricular basal dimension; RVD 2: right ventricular mid-cavity dimension; RVD 3: right ventricular longitudinal dimension; TAPSE: tricuspid annular plane excursion; S wave: peak systolic of the free wall of the right ventricle; PASP: pulmonary artery systolic pressure; PAMP: pulmonary artery mean pressure; PCWP: pulmonary capillary wedge pressure.

**Table 3 jcm-11-07442-t003:** Multiple linear regression analysis of the Δ TR.

Variable	Mean + SD	Β	t	*p*-Value
Δ TR degree at six months	2.5 ± 0.03	-	-	-
Age (years)	59.5 ± 12.9	−0984	0.756	0.082
SBP (mmHg)	108 ± 12.3	−0.063	0.250	0.767
HR (b/m)	62 ± 10.2	−0.265	0.371	0.428
LVEF (%)	30.5 ± 3.6%	0.189	0.465	0.021
PCWP (mmHg)	12.6 ± 5.8	−0543	0.751	0.065
PASP (mmHg)	37.6 ± 8.2	−0345	0.651	0.048
TAPSE (mm)	13.6 ± 5.6	−0012	0.345	0.061

TR: tricuspid regurgitation; SBP: systolic blood pressure; HR: heart rate; LVEF: left ventricular ejection fraction; PCWP: pulmonary capillary wedge pressure; PASP: pulmonary artery systolic pressure; TAPSE: tricuspid annular plane systolic excursion. In bold *p*-value with statistical significance.

## Data Availability

The dataset generated and analyzed in the study is available from the corresponding author on reasonable request.

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
