# Peer review of "Effects of Cardiac Contractility Modulation Electrodes on Tricuspid Regurgitation in Patients with Heart Failure with Reduced Ejection Fraction: A Pilot Study"

_jcm, 2022, doi:10.3390/jcm11247442_

Round 1

Reviewer 1 Report

The manuscript is well written and provides a clear message, i.e. there are no signs of deterioration of tricuspid regurgitation after insertion of electrodes for cardiac contractiliy modulation (CCM). Though the burden of severe tricuspid regurgitation following lead implantation does not appear to be of major clinical significance in clinical routine, these results are helpful with regards to discussions with patients about the safety of the procedure. What may be more interesting and what may warrant further investigations is the observation of an improved pulmonaly artery systolic pressure and an improved tricuspid anular plane excursion that may indicate improved left ventricular filling pressure. Though this might be beyond the scope of the manuscript, the authors may want to consider to eloborate in more detail on the aspect in the discussion section.    

Author Response

The manuscript is well written and provides a clear message, i.e. there are no signs of deterioration of tricuspid regurgitation after insertion of electrodes for cardiac contractiliy modulation (CCM). Though the burden of severe tricuspid regurgitation following lead implantation does not appear to be of major clinical significance in clinical routine, these results are helpful with regards to discussions with patients about the safety of the procedure. What may be more interesting and what may warrant further investigations is the observation of an improved pulmonaly artery systolic pressure and an improved tricuspid anular plane excursion that may indicate improved left ventricular filling pressure. Though this might be beyond the scope of the manuscript, the authors may want to consider to eloborate in more detail on the aspect in the discussion section.    

Response

We thank the reviewer for the appreciation of our paper. Accordingly, to suggestion, we have extended the discussion on the effects of CCM on righty ventricular systolic function and pulmonary pressure.

Reviewer 2 Report

Comment to authors: the authors of this manuscript have in a prospective setting included patients with HFrEF with an indication for OptimizerSmart therapy and evaluated TR regurgitation pre-implant and post-implant as worsening in TR regurgitation is a possibility. This was an important aim as this has not been investigated before. The study included 28 patients with 2 echoes each. The manuscript is well-written though rather short. I have some major comments for the authors:

Why was ICD implantation in the previous 12 months used as an inclusion criteria? Is it a criterira for getting the OptimizerSmart?

How was severe tricuspid regurgitation defined for the exclusion criteria? Same as the methodology described in 1.1 echo evaluation?

In 1.1 echo evaluation the authors write that ROC curves were used for cut-offs – I believe this belongs to the statistics part of the manuscript. Please move.

For the statistics, I assume you used a paired t-test?

Did you consider performing repeated measures multivariable linear regression to control for possible within-person confounders such as hemodynamic variables (e.g. blood pressure at echo scan, heart rate, hgb)?

A possibility is also to do a linear regression with delta_echo measurement and time as the main variables and then inlcude for possible factors that are associated with TR reg worsening? This would add a bit more to your results and could help in understanding which baseline variables are associated with worsening TR regurgitation following OptimizerSmart.

Please include table showing hemodynamic variables at the baseline echo and at the follow-up echo.

A major limitation is the sample size, when looking at your figure 1 it is quite clear that there is a trend in your observations toward increased TR regurgitation as both the VC and PISA increases from baseline to follow-up. When the authors conclude that they did not observe a significant increase in TR regurgitation in the study it is probably mostly due to power. After all, PISA increased 12.5% from baseline to follow-up and similarly for VC value of TR. Please consider rephrasing the conclusion as the conclusion as it is written now is quite “absolute” – the authors write “Implantation of right ventricular electrodes for the delivery of CCM therapy did not worsen TR in patients with HFrEF and previous CIED implantation” which is a bit of a stretch when only 28 patients were examined.

Is there no way for you to group up with another center or similar to include a bit more patients?

In the statistics section, the authors write that inter and intra observer variability was assessed with bland-altman and Pearson’s two-tailed bivariate correlations – I do not see these results reported anywhere in the manuscript?

Author Response

Reviewer 2

Why was ICD implantation in the previous 12 months used as an inclusion criteria? Is it a criterira for getting the OptimizerSmart?

Response: We thank the reviewer for this comment. The implant of transvenous ICD in the previous 12 months was considered exclusion criteria to don’t have confounding factors that could worsen tricuspid regurgitation  

How was severe tricuspid regurgitation defined for the exclusion criteria? Same as the methodology described in 1.1 echo evaluation?

Response: We thank the reviewer for this comment. We have specified the criteria for evaluating severe tricuspid regurgitation in the exclusion criteria section.   

In 1.1 echo evaluation the authors write that ROC curves were used for cut-offs – I believe this belongs to the statistics part of the manuscript. Please move.

Response: We thank the reviewer for this comment. We have corrected it accordingly to the suggestion.

For the statistics, I assume you used a paired t-test?

Response: We thank the reviewer for this comment. We have corrected it accordingly to the suggestion.

Did you consider performing repeated measures of multivariable linear regression to control for possible within-person confounders such as hemodynamic variables (e.g. blood pressure at echo scan, heart rate, hgb)?

A possibility is also to do a linear regression with delta_echo measurement and time as the main variables and then inlcude for possible factors that are associated with TR reg worsening? This would add a bit more to your results and could help in understanding which baseline variables are associated with worsening TR regurgitation following OptimizerSmart.

Response: We thank the reviewer for this comment. However, we believed that the small sample size could affect the result of the linear regression analysis.

Please include table showing hemodynamic variables at the baseline echo and at the follow-up echo.

Response: We thank the reviewer for this comment. We have added some hemodynamic echocardiography-derived parameters to the table 2 .

A major limitation is the sample size, when looking at your figure 1 it is quite clear that there is a trend in your observations toward increased TR regurgitation as both the VC and PISA increases from baseline to follow-up. When the authors conclude that they did not observe a significant increase in TR regurgitation in the study it is probably mostly due to power. After all, PISA increased 12.5% from baseline to follow-up and similarly for VC value of TR. Please consider rephrasing the conclusion as the conclusion as it is written now is quite “absolute” – the authors write “Implantation of right ventricular electrodes for the delivery of CCM therapy did not worsen TR in patients with HFrEF and previous CIED implantation” which is a bit of a stretch when only 28 patients were examined.

Response: Dear reviewer we perfectly agree with your comment, so we have rephrased the conclusion and modified the title of the study.  

Is there no way for you to group up with another center or similar to include a bit more patients?

Response: Unfortunately, we haven’t the possibility to increase the population of the study.

In the statistics section, the authors write that inter and intra observer variability was assessed with bland-altman and Pearson’s two-tailed bivariate correlations – I do not see these results reported anywhere in the manuscript?

Response: We thank the reviewer for this comment. We have added the r coefficient of both tests in the result section.

Round 2

Reviewer 2 Report

The authors have responded to all of my comments. However, they chose not to comply with any of the suggested additional statistical methods to use. I still believe that adding a linear regression model adjusting for the baseline TR parameter value and hemodynamic parameters would improve the results section. It is possible to correct for small sample size in this case. Alternatively a mixed effects repeated measures model could be used in which the degrees of freedom can be handled with a correction for small sample size.

The paper is rather short and the authors have space to expand a bit on their results instead of only using paired t-tests.

Author Response

Dear Reviewer

We have added in the new version of the manuscript, Table 3 with the  multiple linear regression analysis requested and discussed the results of this analisi in the discussion section 

Thanks again for your's useful comments.